# A scoping review on biomedical journal peer review guides for reviewers

Eunhye Song[1], Lin Ang[2,3], Ji-Yeun Park[4], Eun-Young Jun[5], Kyeong Han Kim[6], Jihee Jun[2], Sunju Park[7]*, Myeong Soo Lee[2,3]*

1 Global Strategy Division, Korea Institute of Oriental Medicine, Daejeon, Korea, 2 Clinical Medicine Division, Korea Institute of Oriental Medicine, Daejeon, Korea, 3 Korean Convergence Medicine, University of Science and Technology, Daejeon, Korea, 4 College of Korean Medicine, Daejeon University, Daejeon, Korea, 5 Department of Nursing, Daejeon University, Daejeon, Korea, 6 Department of Preventive Medicine, College of Korean Medicine, Woosuk University, Jeonju, Republic of Korea, 7 Department of Preventive Medicine, College of Korean Medicine, Daejeon University, Daejeon, Korea

* sjpark@dju.kr (SP); drmslee@gmail.com (MSL)

**Data Availability Statement:** All relevant data are within the manuscript and its Supporting Information files.

**Funding:** This work was supported by Korea Institute of Oriental Medicine (KSN2021210 and

## Abstract

### Background

Peer review is widely used in academic fields to assess a manuscript's significance and to improve its quality for publication. This scoping review will assess existing peer review guidelines and/or checklists intended for reviewers of biomedical journals and provide an overview on the review guidelines.

### Methods

PubMed, Embase, and Allied and Complementary Medicine (AMED) databases were searched for review guidelines from the date of inception until February 19, 2021. There was no date restriction nor article type restriction. In addition to the database search, websites of journal publishers and non-publishers were additionally hand-searched.

### Results

Of 14,633 database publication records and 24 website records, 65 publications and 14 websites met inclusion criteria for the review (78 records in total). From the included records, a total of 1,811 checklist items were identified. The items related to Methods, Results, and Discussion were found to be the highly discussed in reviewer guidelines.

### Conclusion

This review identified existing literature on peer review guidelines and provided an overview of the current state of peer review guides. Review guidelines were varying by journals and publishers. This calls for more research to determine the need to use uniform review standards for transparent and standardized peer review.

KSN20212102; awarded to MSL). The funder had no role in study design, data collection and analysis, decision to publish, or preparation of the manuscript.

**Competing interests:** The authors have declared that no competing interests exist.

## Protocol registration

The protocol for this study has been registered at Research Registry (www.researchregistry.com): reviewregistry881.

## 1. Background

In academic fields, peer review is a fundamental part of assessing a manuscript's potential significance to the field and improving its methodological rigor and its overall quality for publishing in scholarly journals. Editors depend on reviewers to make their editorial decisions since the reviewers have expertise in the subject areas of submissions they are reviewing. In terms of writing a paper, there have been many efforts to address inadequate reporting of researches and to achieve the standard in scientific writing through transparent reporting of researches [1]. Besides these guidelines, almost all journals, if not all, provide guidelines for authors at their websites. In recent years, guidelines and checklists have been disseminated by journals and publishers for each of the major study types to ensure consistent and complete reporting [2]. However, guidelines for reviewers are not easily discovered. In a survey of 116 health research journals on the McMaster list, Hirst and Altman found only 41 journals (35%) to be providing online instructions to peer reviewers [3]. Ensuring consistent and complete reviewing is also needed. Publishers like Elsevier, SAGE, Springer Nature, and Wiley provide openly available guidelines for reviewers to assist them through the review process, in order to maintain and support standards in reviewing. Yet, not all publishers provide such guidance to their reviewers openly without any technical barriers such as signing in as a reviewer.

Scoping reviews are similar to systematic reviews in the way they systematically examine evidence and the way they are conducted, but they differ from each other in their purpose. The main differences relate to critical appraisal or risk of bias, results synthesis, and study inclusion. Critical appraisal or risk of bias assessment is necessary for systematic reviews, while it is optional for scoping reviews. Systematic reviews synthesize findings from each included study using meta-analysis or meta-synthesis, while scoping reviews do not synthesize findings [4, 5]. Scoping reviews include more papers broadly to provide an overview of the existing literature regardless of their type or quality [6]. Systematic reviews are used as references for policymaking or clinical practice as they are suitable for clearly defined questions relating to an intervention efficacy, but scoping reviews are descriptive and suitable for examining evidence in a broader sense such as identifying characteristics of the evidence on a topic [7, 8] or identifying available evidence in a field [4]. As peer review is a widely used practice for publication of academic research, there is a need to identify available evidence on peer review guidelines, and to clarify definitions and concepts of peer review process with the evidence.

Currently, there are no widely used standard guidelines for peer reviewing manuscripts which may intimidate or frustrate inexperienced peer reviewers [9]. A lack of standardization may lead to difficulty in reviewing as they are uncertain about what are expected from them. Little has been published about peer review and not much resources have been provided [10]. This scoping review will determine existing peer review guidelines and/or checklists intended for reviewers of biomedical journals. The purpose of this study is to determine whether there are enough evidence and research to set standards on how reviewers of biomedical journals should perform peer reviews and write their review reports, and to provide an overview on the review guidelines.

## 2. Materials and methods

A scoping review of manuscript peer review guidelines for reviewers of biomedical journals was conducted to identify available evidence on peer review guidelines for reviewers and to determine what is communicated to peer reviewers about how to peer review. In order to improve the methodological and reporting quality of this scoping review, the checklist of PRISMA Extension for Scoping Reviews (PRISMA-ScR) [7], developed in 2018 by the EQUATOR Network was used (S1 File).

### 2.1. Registration

The protocol for this study has been registered at Research Registry with its unique identifying number, reviewregistry881 in April 2020.

### 2.2. Databases and search strategy

PudMed, Embase, and Allied and Complementary Medicine Database (AMED) were all searched from the date of inception until February 19, 2021. In the search, there was no date restriction. Reviewer guidelines by some of the major publishers, journals, and non-publishers were additionally hand-searched online and collected.

For comprehensive search on peer review guidelines in biomedical field, the following search terms were used: "reviewing", "reviewer", "peer review", "peer reviewer", "referee", "journal", "manuscript", "guide", "guidelines", "how to", "lesson", "tips", "advice", "recommendation", "report form", and "checklist". In order to search more publications for comprehensive results, Text Word search was performed, and MeSH term for "peer review" was used for search in PubMed. The search strategy was reviewed by all authors and agreed upon. The complete search strategy can be found in S2 File.

For grey literature, websites of major biomedical journal publishers were searched manually by browsing and clicking through the websites to look for guidelines for reviewers. The publishers were selected based on Simba Information (www.simbainformation.com)'s report, Global Medical Publishing 2019–2023, which provides a list of the leading medical publishers. More publishers were selected from those on Journal Citation Report (jcr.clarivate.com) in the Integrative and Complementary Medicine category. Subsidiary websites were not included in the search of major publishers. The websites searched are as follows: Elsevier (www.elsevier.com), Springer (www.springer.com), Nature (www.nature.com), Wiley (www.wiley.com), Wolters Kluwer (www.wolterskluwer.com), Taylor and Francis (www.tandfonline.com), SAGE (www.sagepub.com), PLOS (www.plos.org), MDPI (www.mdpi.com), Hindawi (www.hindawi.com), Thieme (www.thieme.com), Mary Ann Liebert (www.liebertpub.com), Cognizant Communication Corporation (cognizantcommunication.com), InnoVision Health Media (innovisionhm.com), World Scientific Publishing (www.worldscientific.com), BioMed Central (www.biomedcentral.com), Karger Publishers (www.karger.com), and Lippincott Williams and Wilkins (lww.com). In addition, the websites of non-publishers were searched including Council of Science Editors (www.councilscienceeditors.org), the Committee on Publication Ethics (www.publicationethics.org), the EQUATOR Network (www.equator-network.org), and the ICMJE (www.icmje.org), the World Association of Medical Editors (www.wame.org), the International Congress on Peer Review and Scientific Publication (peer-reviewcongress.org) and Publons (www.publons.com). For the EQUATOR reporting checklists published both on their website and some journals, they were considered as website items all under the EQUATOR Network as a non-publisher organization.

## 2.3. Inclusion and exclusion criteria

The inclusion criteria for this study were any publication relating to reviewer guidelines or checklists that a manuscript reviewer may refer to point by point during peer review, websites of publishers and non-publisher organizations to determine the evidence within the biomedicine field. All types of articles in English (or English translation) were eligible for inclusion in this study without restrictions on date or study design. Thus, editorial materials such as editorial and tutorials were included. Among the websites of above-mentioned publishers and non-publisher organizations in the 2.2 Databases and search strategy section, those with reviewer guidelines or checklists were eligible for inclusion.

The exclusion criteria were statistical review, conference abstracts, non-English articles, checklists primarily for authors, review process related contents, reviewer/editor roles and duties, assessing review quality or reliability, review challenges or issues, conference peer reviews, grant peer reviews, and those not relating to manuscript peer review. Websites that do not provide reviewer guidelines were not eligible for inclusion.

In order to assess and organize the included publications, a modified quality assessment tool was modified from the tool suggested by Hawker et al. [11] and used in this study to assess the publications included.

## 2.4. Outcomes

The primary outcomes were review checklist items such as a list of questions or points to consider during peer review. The secondary outcomes were the review guidance criteria. Before going through the screening, after the search of literature, the authors randomly pre-screened 10 publications to look for possible review guidance criteria, and based on the discussion by all authors with experience as peer reviewers, the criteria were developed. These criteria aimed to group the contents of reviewer guidelines, such as instructions or recommendations on the reviewer's attitude, becoming a reviewer, manuscript contents, review resources, reviewer ethics, manuscript structure, study ethics, time management, and other components relating to reviewing manuscripts, and to provide a clear overview of available guidelines for reviewers. Then for each included publication, the criteria were independently selected by two authors.

## Glossary 1. Description of each criterion

| Criterion | Description |
|---|---|
| Attitude | guidelines on how to behave toward authors or how to keep the positive and constructive tone |
| Becoming a Reviewer | guidelines on how to become a reviewer |
| Contents | guidelines on how to assess manuscript contents like introduction, methodology, results, discussion, and conclusion |
| Resources | guidelines on how to access or refer to some resources related to reviewing |
| Reviewer Ethics | guidelines on how to deal with any possible reviewer ethical issues |
| Structure | guidelines on how to organize or write the reviewer comments |
| Study Ethics | guidelines on how to deal with ethical concerns found in manuscripts that the reviewers review |
| Time Management | guidelines on how to manage or deal with reviewing time |
| Other | guidelines on other components relating to manuscript peer review, including signing reviews for increased transparency, reviewer recognition, and collaborating with others |

## 2.5. Study selection and data extraction

A data extraction form was developed for this scoping review for data charting purposes. The data extracted from the included articles were first author's last name, publication year, article type, guidance criteria, and number of checklist items.

Two independent review authors searched the databases and assessed the eligibility of the searched records. Any discrepancies on article inclusion were discussed with a third review author and resolved on consensus.

All checklist items for reviewing manuscripts were extracted into an Excel spreadsheet. Each item was identified and categorized by its relevant section in manuscript review. The section categories used were title, abstract, introduction, methods, results, discussion, conclusion, references, tables and figures, presentations, and other. The categories were based on the general sections and components of manuscripts and also modified from the category structure of the PRISMA 2009 checklist [12]. The items from the included records of publications and websites were categorized first by the original category of the checklist if they match. For instance, for the checklist items that were categorized under the methods category on the checklist, we categorized it accordingly under the methods category. The items without any specific category were reviewed by the two authors and their categories were determined.

# 3. Results

## 3.1. Search results

After screening and excluding articles and other records that fell under exclusion criteria, a total of 79 records (65 publications on peer review guidelines and 14 publisher and non-publisher websites with reviewer guidelines) were included for this study (Fig 1). Out of 79 records, 24 publications did not provide a checklist as in what to look for in the manuscript (relating to contents) and only provided guidance for reviewers as to how to review. The 65 included publications and the 14 websites contained 1,077 items and 734 items respectively (S3 File), which amounted to 1,811 items in total. Each item is a question or points to consider, such as "Is the title appropriate?" or "Accurate/Adequate Abstract." The assessment results of each included publication are provided as a supplement (S4 File).

## 3.2. Characteristics of included articles and websites

The descriptive characteristics of the included articles and websites are summarized in Tables 1 and 2 respectively. Publication types of the included studies were mostly editorial materials (n = 48; 75.0%) such as tutorial, editorial, special articles, perspective, opinion, and commentary. Other types included were review articles (n = 14; 21.9%), research articles (n = 2; 3.1%), and guidelines (n = 1; 1.6%). These article types were extracted as they were presented in the publications. Among the included publications that provided information on how to peer review, 41 publications (63.1%) provided a checklist questions or criteria, but 24 publications (36.9%) did not provide a checklist questions or criteria.

## 3.3. Criteria of reviewer guidelines

The suggestions and instructions for reviewers provided in the included publications were grouped into 9 criteria: attitude, becoming a reviewer, contents, resources, reviewer ethics, structure, study ethics, time management, and other as presented in 2.4 Outcomes in the Methods section. Publications were determined to fit into one or more of the criteria as appropriate. The frequency of reviewer guides criteria extracted from the publications is provided at the bottom rows in Table 1.

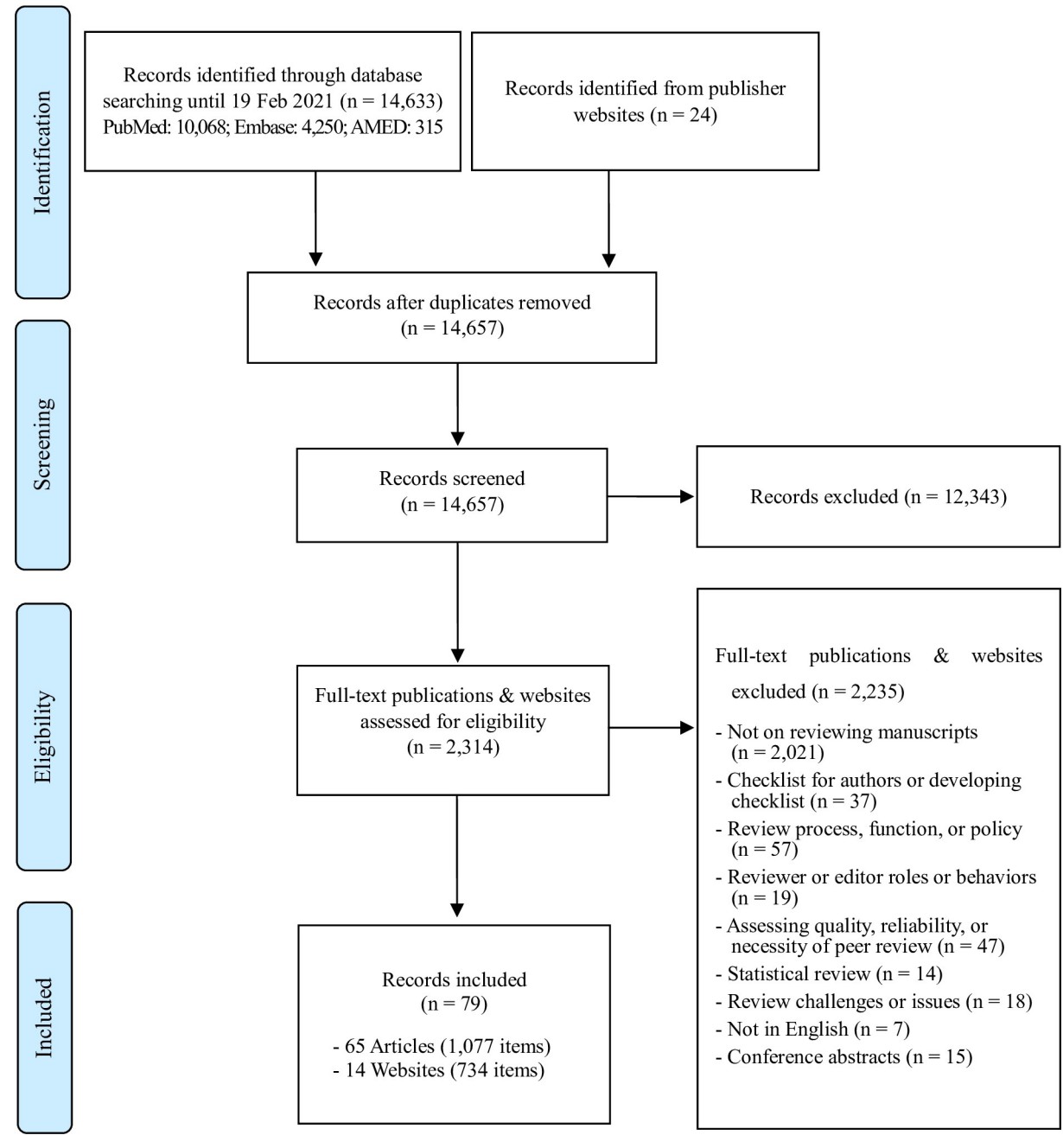

**Fig 1. Flow diagram for the review.**

## 3.4. Categories of reviewer checklist items

The checklist items were extracted from the checklists in the included publications and websites, and the items were grouped into 17 categories (Table 3). The top 3 categories with the largest number of items were the Methods (548 items, 30.3%) category, the Results (231 items, 12.8%) category, and the Discussion (191 items, 10.5%) category from the publication items and website items combined. Looking into them separately by publications and websites, the

**Table 1. Characteristics of included publications–database search results (65 publications).**

| Author & Year | Publication Type | Guidance Criteria[#] | | | | | | | | | No. of Checklist Items |
| --- | --- | --- | --- | --- | --- | --- | --- | --- | --- | --- | --- |
| | | Attitude | Becoming a Reviewer | Contents | Resources | Reviewer Ethics | Structure | Study Ethics | Time Management | Other | |
| Ades, 2013 [13] | Tutorial | | | ✓ | | | | | | | 42 |
| Alam, 2015 [2] | Editorial | ✓ | | ✓ | ✓ | | ✓ | | ✓ | | NP |
| Alexander, 2005 [14] | Editorial | | | ✓ | | | ✓ | | | | 19 |
| Alexandrov, 2009 [15] | Special Article | ✓ | | ✓ | | ✓ | ✓ | | ✓ | | 7 |
| Allen, 2017 [16] | Editor's Perspective | ✓ | | ✓ | ✓ | | ✓ | | ✓ | | 10 |
| Allen, 2014 [17] | Scientific Perspectives | ✓ | | ✓ | | ✓ | ✓ | | ✓ | | 21 |
| Altieri, 2020 [18] | Mentor of the Month | | ✓ | ✓ | ✓ | | ✓ | | | | NP |
| Annesley, 2013 [19] | Tutorial* | ✓ | | ✓ | | | ✓ | | | | NP |
| Benos, 2003 [20] | A Personal View | ✓ | | ✓ | ✓ | ✓ | | ✓ | ✓ | | 14 |
| Brand, 2012 [21] | Editorial | ✓ | | ✓ | | ✓ | ✓ | | | | 31 |
| Brown, 2017 [22] | Opinion | | | ✓ | | | ✓ | | ✓ | ✓ | 22 |
| Cantor, 2009 [23] | Editorial | | ✓ | ✓ | | ✓ | ✓ | | | | NP |
| Christenbery, 2011 [24] | Education | ✓ | | ✓ | | ✓ | | ✓ | ✓ | ✓ | 36 |
| Crigger, 1998 [25] | Editorial | ✓ | | | | ✓ | | | ✓ | | NP |
| Currie, 2016 [26] | Special Contribution | | | ✓ | | ✓ | ✓ | ✓ | ✓ | | NP |
| Del Mar, 2015 [27] | Tutorial | | | ✓ | | ✓ | ✓ | ✓ | ✓ | | NP |
| Dhillon, 2021 [28] | Words of Advice | ✓ | | ✓ | ✓ | ✓ | ✓ | | ✓ | | NP |
| Duchesne, 2008 [29] | Research | | | ✓ | | | | | | | 70 |
| Duff, 2009 [30] | Commentary | ✓ | ✓ | ✓ | | ✓ | ✓ | | ✓ | ✓ | NP |
| Einarson, 2012 [31] | Review* | ✓ | | ✓ | | | ✓ | | | | NP |
| England, 2019 [32] | Review | | ✓ | | | | | | | | 14 |
| Estrada, 2006 [33] | Editorial* | ✓ | | | | | ✓ | | ✓ | | 11 |
| Genter, 2020 [34] | Editor's Tips | ✓ | | ✓ | | ✓ | ✓ | | ✓ | | NP |
| Halder, 2011 [35] | Review | ✓ | ✓ | ✓ | ✓ | ✓ | ✓ | | ✓ | | NP |
| Heddle, 2009 [36] | Clinical Research Focus | ✓ | | ✓ | | ✓ | ✓ | | ✓ | | 16 |
| Hill, 2016 [37] | Editorial* | ✓ | | ✓ | | ✓ | ✓ | | | | 8 |
| Hunter, 2020 [38] | Research | | | ✓ | | | | | | | 15 |
| Kelly, 2014 [39] | Review* | ✓ | | ✓ | | | ✓ | | ✓ | | NP |
| Kocak, 2020 [40] | Review | | | ✓ | | | | | | | NP |
| Kotsis, 2014 [41] | Review* | | | | | ✓ | ✓ | | | | 37 |
| Kottner, 2016 [42] | Editorial | ✓ | | ✓ | ✓ | ✓ | ✓ | | | | NP |
| Kyrgidis, 2010 [43] | Review | | | ✓ | ✓ | ✓ | | | | | 16 |
| Lapin, 2020 [44] | Review | | | ✓ | | | | | | | 9 |
| Lazarides, 2020 [45] | Review | ✓ | | ✓ | | ✓ | ✓ | | ✓ | | NP |

(*Continued*)

**Table 1.** (Continued)

| Author & Year | Publication Type | Guidance Criteria[#] | | | | | | | | | No. of Checklist Items |
| --- | --- | --- | --- | --- | --- | --- | --- | --- | --- | --- | --- |
| | | Attitude | Becoming a Reviewer | Contents | Resources | Reviewer Ethics | Structure | Study Ethics | Time Management | Other | |
| Lippi, 2018 [46] | Perspective | ✓ | | ✓ | | ✓ | ✓ | | ✓ | | 20 |
| Marusic, 2005 [47] | Guidelines | ✓ | | ✓ | | ✓ | ✓ | | ✓ | | 23 |
| Moher, 2015 [48] | Tutorial | ✓ | | ✓ | | | ✓ | | | | NP |
| Oerther, 2019 [49] | Editorial | ✓ | ✓ | ✓ | | ✓ | ✓ | | ✓ | | 31 |
| Pai, 2020 [50] | Perspective | ✓ | | ✓ | | ✓ | | | ✓ | | 46 |
| Paice, 2001 [51] | Medical Publishing Series | | ✓ | ✓ | | ✓ | ✓ | | ✓ | | 13 |
| Pietrzak, 2010 [52] | Special Editorial | | ✓ | ✓ | | ✓ | ✓ | | ✓ | | NP |
| Provenzale, 2005 [53] | Perspective | ✓ | | ✓ | | ✓ | | | | | 35 |
| Rosenfeld, 2010 [54] | Special Article | ✓ | | ✓ | | ✓ | ✓ | | ✓ | | 57 |
| Rostami, 2011 [55] | Medical Education | | | ✓ | ✓ | ✓ | ✓ | | ✓ | | 18 |
| Rutkowski, 2009 [56] | Editorial | ✓ | | ✓ | | ✓ | ✓ | | ✓ | | NP |
| Salasche, 1997 [57] | Editorial | | ✓ | ✓ | | ✓ | ✓ | | ✓ | | NP |
| Sasson, 2021 [58] | Review | | ✓ | ✓ | | ✓ | ✓ | | ✓ | | NP |
| Schuttpelz-Brauns, 2010 [59] | Position paper | ✓ | | ✓ | | | ✓ | | | | 9 |
| Seals, 2000 [60] | Innovation and Ideas | | | ✓ | | | ✓ | | | | 54 |
| Simpson, 2008 [61] | Tutorial* | ✓ | | ✓ | | | ✓ | | ✓ | | 37 |
| Small, 2019 [9] | Editor's Corner | | ✓ | ✓ | ✓ | ✓ | ✓ | | | | NP |
| Smith, 2019 [62] | Review | | | ✓ | | | ✓ | | ✓ | | 34 |
| Smolcic, 2014 [63] | Research Integrity Corner | ✓ | | ✓ | | ✓ | ✓ | | ✓ | | 65 |
| Son, 2021 [64] | Editorial | ✓ | ✓ | ✓ | | | | | | | NP |
| Stahel, 2016 [65] | Tutorial | ✓ | | ✓ | ✓ | | ✓ | | ✓ | ✓ | 18 |
| Stenfors, 2020 [66] | How To | | | ✓ | | | | | | | 5 |
| Stone, 2018 [67] | Editorial | | | ✓ | | | ✓ | | | | 9 |
| Sucato, 2018 [68] | Review | ✓ | | ✓ | ✓ | ✓ | ✓ | | ✓ | | 40 |
| Sylvia, 2001 [69] | Special article | ✓ | ✓ | ✓ | ✓ | ✓ | ✓ | | | | 13 |
| Talanow, 2014 [70] | Editorial | | | ✓ | | | | | | | 10 |
| Tandon, 2014 [71] | Review | | | ✓ | | | ✓ | | ✓ | | 37 |
| Tullu, 2020 [72] | Editorial | ✓ | ✓ | ✓ | ✓ | ✓ | ✓ | ✓ | ✓ | | 54 |
| Venne, 2015 [73] | Professional Issues | ✓ | | ✓ | | ✓ | ✓ | | ✓ | | NP |
| Walker, 1997 [74] | Tutorial* | | | ✓ | | | | | | | 40 |
| Wilson, 2002 [75] | Review* | | | ✓ | | | | | | | 11 |

(*Continued*)

**Table 1.** (Continued)

| Author & Year | Publication Type | Guidance Criteria# | | | | | | | | | No. of Checklist Items |
| | | Attitude | Becoming a Reviewer | Contents | Resources | Reviewer Ethics | Structure | Study Ethics | Time Management | Other | |
|---|---|---|---|---|---|---|---|---|---|---|---|
| Total guidance criteria | n | 37 | 14 | 61 | 14 | 37 | 48 | 5 | 37 | 4 | |
| | % | 57.8 | 21.9 | 95.3 | 21.9 | 57.8 | 75.0 | 7.8 | 57.8 | 6.3 | |

# The criteria refer to, Attitude: how to behave toward authors or how to keep the positive and constructive tone; Becoming a Reviewer: how to become a reviewer; Contents: how to assess manuscript contents like introduction, methodology, results, discussion, and conclusion; Resources: how to access or refer to some resources related to reviewing; Reviewer Ethics: how to deal with any possible reviewer ethical issues; Structure: how to organize or write the reviewer comments; Study Ethics: how to deal with ethical concerns found in manuscripts that the reviewers review; Time Management: how to manage or deal with reviewing time; Other: other guidance including signing reviews for increased transparency, reviewer recognition, and collaborating with others.

* Article type not reported; assumed article type added.

top 3 categories with the most items were also the same 3 categories of Methods, Results, and Discussion.

The grouped categories of the identified checklist items and their numbers are shown in a graphical representation (Fig 2). In the figure, the darkness of color and the size of diamond shape are associated with the number of items for each category.

## Discussion

From an extensive range of sources, this scoping review produced a comprehensive list of items outlining guidance for conducting peer review. Review guidelines and checklists can be important for high quality review, especially since editors rely on reviewers and peer reviewer recommendations have an influence on editorial decisions [76]. Such influence of reviewer recommendations on editorial decisions becomes an issue when the quality of peer review is doubtful or when reviewer bias is present [77]. In order to prevent reviewer bias and better

**Table 2. Characteristics of included websites–hand search results (14 websites, 734 items).**

| Website Source | Organization Type* | No. of Checklist Items |
|---|---|---|
| Council of Science Editors | Non-publisher | 76 |
| EQUATOR Network** | Non-publisher | 378 |
| ICMJE | Non-publisher | 67 |
| Publons | Non-publisher | 18 |
| Elsevier | Publisher | 29 |
| Hindawi | Publisher | 14 |
| MDPI | Publisher | 7 |
| Nature | Publisher | 10 |
| PLOS | Publisher | 27 |
| SAGE | Publisher | 15 |
| Springer | Publisher | 7 |
| Taylor and Francis | Publisher | 64 |
| Wiley | Publisher | 14 |
| Wolters Kluwer Medicine | Publisher | 8 |

* Non-publisher organizations include council, network, committee, and company.

** The reporting guidelines for main study types were searched: CONSORT, STROBE, PRISMA, SPIRIT, STARD, CARE, SQUIRE, AGREE, ARRIVE, CHEERS, and SRQR.

**Table 3. Extracted checklist items by categories (1,811 items; 17 categories).**

| Category | Publication Items | | Website Items | | Total | |
|---|---|---|---|---|---|---|
| | n | % | n | % | n | % |
| Title | 23 | 2.1 | 18 | 2.5 | 41 | 2.3 |
| Abstract | 52 | 4.8 | 37 | 5.0 | 89 | 4.9 |
| Keywords | 5 | 0.5 | 2 | 0.3 | 7 | 0.4 |
| Introduction | 90 | 8.4 | 49 | 6.7 | 139 | 7.7 |
| **Methods** | **311** | 28.9 | **237** | 32.3 | **548** | 30.3 |
| **Results** | **131** | 12.2 | **100** | 13.6 | **231** | 12.8 |
| **Discussion** | **124** | 11.5 | **67** | 9.1 | **191** | 10.5 |
| Conclusion | 35 | 3.2 | 25 | 3.4 | 60 | 3.3 |
| References | 47 | 4.4 | 25 | 3.4 | 72 | 4.0 |
| Statistics | 30 | 2.8 | 19 | 2.6 | 49 | 2.7 |
| Tables/Figures | 64 | 5.9 | 37 | 5.0 | 101 | 5.6 |
| Ethical Concerns | 45 | 4.2 | 42 | 5.7 | 87 | 4.8 |
| Significance/Relevance | 47 | 4.4 | 21 | 2.9 | 68 | 3.8 |
| Originality | 17 | 1.6 | 5 | 0.7 | 22 | 1.2 |
| Reporting | 4 | 0.4 | 1 | 0.1 | 5 | 0.3 |
| Presentation | 43 | 4.0 | 27 | 3.7 | 70 | 3.9 |
| Other* | 9 | 0.8 | 22 | 3.0 | 31 | 1.7 |
| **Total** | **1,077** | **100.0** | **734** | **100.0** | **1,811** | **100.0** |

* Other: Accessibility, Article Category, Data Availability, Journal Scope, Reviewer Expertise, Software Availability, and Terminology.

Bold values indicate top 3 categories among Publications, Websites, and Total (Publications & Websites) items.

improve the peer review quality, some journals and publishers published and posted reviewer guidelines. However, Davis et al. [78] found the content of the manuscript review evaluation forms to vary considerably and relatively few journals asked reviewers to rate specific components of a manuscript. In their study, the review evaluation form of higher impact factor journals addressed statistical analysis, ethical considerations, and conflict of interest, while lower impact factor journals addressed scientific importance, validity, and novelty. Similarly, Gardner et al. [79] also stated that a reviewer report form was found to be more important in reviewing Methods and Results sections in manuscripts. In addition, in a study by Strayhorn et al. [80], the use of standardized review checklist was found to increase reviewer agreement.

In this review, the review guidance criteria with the two highest frequencies were found to be the Contents (95.3%) criterion and the Structure (75.0%) criterion. These are considered important components to consider during peer review and many of the included publications were highly focused on them. They reflect the quality of the peer review report inside (reviewing on manuscript contents) and outside (organizing review report structure). This is because the main functions of peer review are to assess the manuscript contents and to have the review results clearly delivered to editors and authors. In terms of the manuscript categories among the checklists, the checklist items with the most in number were the Methods, Results, and Discussion categories. This indicates that the authors as well as publishers and non-publisher organizations consider these categories to be the main parts of manuscripts that require careful evaluation over other parts as the manuscript quality depends highly on these parts when reviewed properly. However, bias frequently occurred while reviewing them which eventually affects editors and editorial decisions [81, 82]. The quality of the manuscripts could be improved by providing reviewer guidelines, especially touching on such important components. Although the reporting guidelines by the EQUATOR network (www.equator-network.

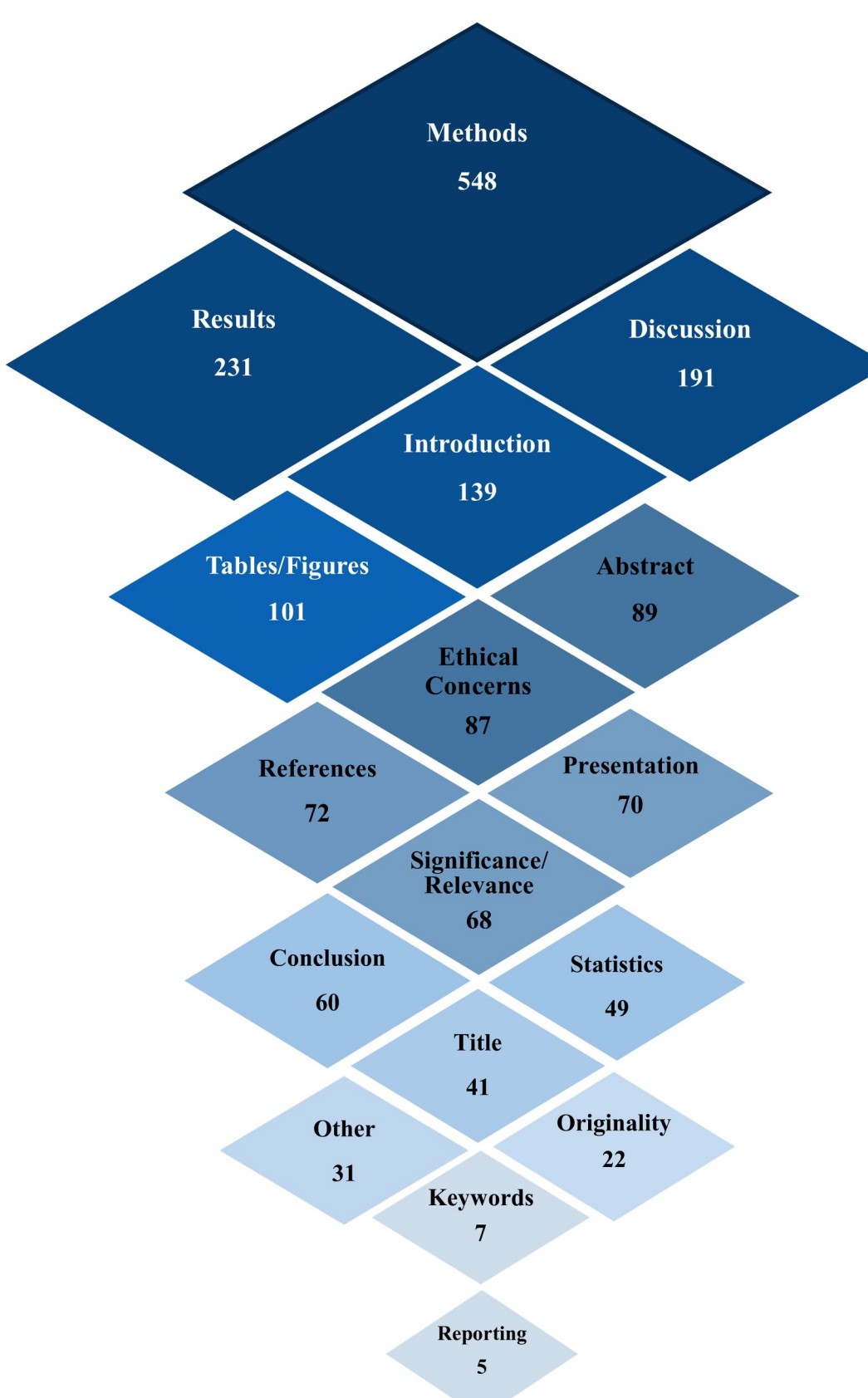

**Fig 2. Graphical representation of the review component categories extracted from the publications and websites (1,811 items; 17 categories).**

org) are now widely accepted by major publishers and journals, they are most commonly used by authors in their reporting, since the main purpose of those guidelines was to improve author reporting. For this reason, guidelines by publishers or journals placed much less significance on guidelines for reviewers. Yet, reviewers are expected to cover a wide range of tasks. Glonti et al. [77] conducted a scoping review on the roles and tasks of peer reviewers and found reviewer tasks to include: organizing reviewing, making general comments, assessing and addressing content for each section of the manuscript, addressing ethical aspects, assessing manuscript presentation and providing recommendations. Manuscript peer reviews could be improved by increasing the consistency of ratings, narrative comments, and recommendations [83]. In order to improve the reliability of reviewers, one of the ways suggested by Kravitz et al. [84] was providing more effective guidelines for reviewers, and one suggested by Kwon et al. [85] was providing a checklist to reviewers. In more recent study, Stenfors et al. [66] also suggested that guidelines may help to better identify criteria indicating high-quality research in journal publications. DiDomenico et al. [86] discussed about previous studies that investigated the effectiveness of formal training on review quality and found the training to influence the review quality insignificantly or limitedly; thus, the authors suggested that reviewers may benefit from some guidance.

Primarily, checklists are meant to be mnemonic devices aimed at supporting authors and reviewers through a process with a general outline. The items for the evaluation of the manuscript structure or format had more overlaps than those for the evaluation of the manuscript content such as methods and results. The general items of the review checklist in the biomedical related field are most likely to overlap with non-biomedical field as well. As the checklist can be designed based on the overall purpose, some checklist can be more useful than others. Therefore, in order to improve current peer review conditions, we suggest that more research should be carried out and the various groups involved should come together to discuss about the need of a more standardized checklist to be developed in a systematic format and useful to many groups. This kind of attempt is still in progress as seen in a study protocol of randomized controlled trial by Speich et al. [87]. This trial aims to evaluate the effectiveness or the need of providing a short version of the CONSORT checklist and its explanation to peer reviewers for them to check during peer review.

Getting the various groups involved and developing a more standardized checklist would have a technical merit. The items employed for a checklist should be from a broad range of academic groups and reliable sources. Various perspectives that could represent the reviewer committee in general should be considered and factored into the design of the checklist. This is particularly important if the checklist is designed to decrease errors of omission during peer review. Although reviewer checklists themselves are generally not included for publication even for open peer review, the development of reviewer guidelines should be evidence based and requires a systematic and comprehensive approach. The design of effective checklist should include utilization of published guidelines, formation of expert panels for consensus and extensive pilot-testing of preliminary checklists to ensure that recommendations are valid, reliable, and applicable.

These days, being transparent and open about peer review process has gained more attention and is a critical issue for publishers and editors because it fosters them to reach fair and consistent decisions [88], which would then facilitate the authors to improve their manuscript and to move forward. Having standards in peer reviewing could make reviewers more at ease

and simplify their review. As the guidelines for author reporting has encouraged structure standardization for publications, a uniform review guideline checklist would support the transparent peer review to be more transparent and standardized for the journals in the same field with similar scope. It would also help authors to better understand their insufficient parts of the manuscript. However, more evidence is needed to evaluate the effects of peer review checklist. In addition, on top of the general standardized review checklist, a list of specific questions that are tailored for the journal, depending on the study design or topic, could be added on.

Limitation of this study was the inclusion of mainly editorial publication types. Most of the included publications being editorial materials places concerns for the guidelines and check-lists suggested, since editorials present the perspective of only the editorial's author. Scoping reviews cover a wide range of publication types that quality assessment is not mandatory to perform [4, 6]. The publications on "how to review" were mostly editorial materials, including editorials, tutorials, and special contributions, and the generally used quality assessment tools could not be applied to them so the modified quality assessment was used for the assessment of the included publications. In addition, there were limitations in extracting more grey litera-ture. Many journals do not provide reviewer checklists openly and they cannot be accessed by anyone. Thus, the checklists openly provided by publishers were only considered. While the manual search of websites for checklists added more to our data, this may have introduced human error. Other limitations were bias related to search strategies and inclusion criteria. Our preselected keyword search strategies may have introduced some bias or unintended exclusion of publications which should have been included. In addition, biomedical databases with access restrictions were also searched to provide more comprehensive data but this may cause barriers to the publications for researchers. The inclusion of only English language pub-lications may have also excluded some content.

## 5. Conclusion

This scoping review explored peer reviewer guidelines in biomedical journals. It identified the extent of existing literature on peer review guidelines and provided an overview of peer review guidelines. It demonstrated the current state of peer review guidelines that they were not uni-form which may imply the need for more uniform guidelines. This review could provide grounds for future studies related to reviewer guidelines. In order to produce objective criteria for reviewing manuscripts, further research is needed, and various groups involved in publish-ing academic manuscripts, such as publishers, editors, reviewers, and other experts, should come together to discuss on developing standard guidelines or recommendations for peer reviewers which may be used by itself or along with individual journal's instructions for reviewers.

## Supporting information

**S1 File. Preferred Rreporting Items for Systematic reviews and Meta-Analyses extension for Scoping Reviews (PRISMA-ScR) checklist.**
(DOCX)

**S2 File. Search strategy.**
(DOCX)

**S3 File. Included review checklist items.**
(DOCX)

**S4 File. Quality assessment of included publications (n = 65).**
(DOCX)

## Author Contributions

**Conceptualization:** Sunju Park, Myeong Soo Lee.

**Formal analysis:** Eunhye Song, Lin Ang, Jihee Jun.

**Funding acquisition:** Myeong Soo Lee.

**Investigation:** Eunhye Song, Lin Ang.

**Writing – original draft:** Eunhye Song.

**Writing – review & editing:** Ji-Yeun Park, Eun-Young Jun, Kyeong Han Kim, Sunju Park, Myeong Soo Lee.

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
