## [Decision Letter · Decision Letter 0]

2 Feb 2021

PONE-D-20-36943

A scoping review on biomedical journal peer review guides for reviewers

PLOS ONE

Dear Dr. Ang,

Thank you for submitting your manuscript to PLOS ONE. After careful consideration, we feel that it has merit but does not fully meet PLOS ONE’s publication criteria as it currently stands. Therefore, we invite you to submit a revised version of the manuscript that addresses the points raised during the review process.

We look forward to receiving your revised manuscript.

Kind regards,

Tim Mathes

Academic Editor

PLOS ONE

Journal Requirements:

2. Please amend the manuscript submission data (via Edit Submission) to include authors Ji-Yeun Park, Eun-Young Jun, Kyeong Han Kim, Jihee Jun, Sunju Park, and Myeong Soo Lee.

Reviewers' comments:

Reviewer's Responses to Questions

**Comments to the Author**

1. Is the manuscript technically sound, and do the data support the conclusions?

Reviewer #1: Partly

Reviewer #2: Partly

2. Has the statistical analysis been performed appropriately and rigorously? 

Reviewer #1: Yes

Reviewer #2: N/A

3. Have the authors made all data underlying the findings in their manuscript fully available?

Reviewer #1: Yes

Reviewer #2: No

4. Is the manuscript presented in an intelligible fashion and written in standard English?

Reviewer #1: Yes

Reviewer #2: No

5. Review Comments to the Author

Reviewer #1: Review comments on “A scoping review on biomedical journal peer review guides for reviewers”

Thank you for allowing me the privilege to read your manuscript and provide feedback on your work. All of the comments I respectfully submit with the aim to make your article stronger and more easily read and understood by the community of readers who will find it at PLOS ONE.

The attached pdf file of your article includes comments and highlights specifically outlining changes that I think will improve your argument and the article as a whole.

Is the manuscript technically sound, and do the data support the conclusions?

I have answered “partly” to this question. There are several small changes you can make to the manuscript that will strengthen it. First, you make claims in your discussion section that I do not feel are warranted by the data you collected. It is not that these claims should not be considered, rather they simply need more research before we can make them. I suggest moving these to an expanded section discussing areas for further research.

It will also strengthen your article if you can provide more explicit scoping for the inclusion and exclusion criteria that you used in your study. For example, please clarify why you chose the websites that you listed, and why not other websites? (e.g. F1000Research, Peer J, eLife.) Additionally, if you can provide more detail about how those websites were searched.

Your search strategies used are decent, though I do have questions about some of the searching decisions you made. Please make clear why you made certain decisions to search as you did and include that in your methods section.

It would also strengthen your article if you expand upon the limitations of your study. I have outlined potential additions to the limitations on the uploaded pdf in the comments and changes. One example is that you might refer to the search strategy as a limitation, as well as the inclusion of articles only written in English.

Another way that you could strengthen this article would be to expand the conclusion and mention more specifically what other studies may come of your findings.

There is also one more substantive change to consider—the nomenclature used to discuss articles. It may be much clearer to your readers if you refer to the data retrieved from databases as “publications” instead of “articles.” Given that “articles” are often used by databases as a publication type, using this word in the text of your manuscript to discuss the data can introduce confusion. If you were to use the word “publication” to mean those items found in databases, some of which *are* research articles, this would help eliminate some confusion.

Has the statistical analysis been performed appropriately and rigorously?

Your data analysis is thorough.

Have the authors made all data underlying the findings in their manuscript fully available?

Thank you for using the registry as well as the PRISMA-ScR.

Supplemental files and other figures made my job as a referee much easier, and will help other researchers develop subsequent studies to further knowledge in this area.

Is the manuscript presented in an intelligible fashion and written in standard English?

I answered yes to this question, though I have made a few recommendations to make your writing even clearer.

Concluding remarks

There are a number of minor changes that I think will greatly strengthen this article. As I have noted on the manuscript itself and in this document, please provide more details in your methods section regarding inclusion/exclusion, search strategies, etc. Please also expand the limitations section and areas for further study. Finally, please move conclusions unsubstantiated by the data into the areas for further study.

Thank you again for allowing me to review your work.

Respectfully submitted by Emily Ford

1/9/2021

Reviewer #2: Song and colleagues address a critical issue in medical scholarship ecosystem, namely, the role of peer review. Their scoping review addresses a practical point about the availability of peer review guidance/checklist for manuscripts. The scoping review identified a large number of reports that met the scoping reviewer’s inclusion criteria. The authors provide some information about the included documents. While I think the scoping review provides some utility for readers, I think it needs more rigor and interpretation before it can be more meaningful to readers.

I am delighted that the authors registered their scoping review and used PRISMA-ScR to aim in reporting their scoping review.

A main concern of mine is with the output of the results. While the authors provide a long list of peer review guidance (Table 1), as a reader I’m lost. What usefulness to readers is an editorial with guidance criteria A,C,M,R,S (e.g., Alam, 2015)? If I go to Figure 2, I can try and unpack what this means. For example, I can see that A is ‘attitude’. But from a reader’s point of view what does this mean for peer reviewing? One way of solving this issue is to include a box with a clear glossary of explanation for each criterion in Figure 2 – this information should be included in the Methods section. The authors also need to describe how these criteria were selected and developed – please include this in the methods section as well.

The other aspect of Table 1 that concerns me is that it appears each row is equivalent to each other. There is no evidence-based hierarchy. For example, is an editorial with no checklist items as informative/evidence based as a guideline (e.g., Marusic, 2005). Should there no be a vetting and organization of all the guidance documents included? For this reason, I do wonder whether there is a need for the authors to complete some form of validity assessment of all of the included documents. You as authors, and readers more generally, need some way of separating the wheat from the chaff. The results of the scoping review are also silent on whether there is overlap between the guidance documents. This point relates to the discussion. I have included a recent systematic review, also about checklists. The authors might find it helpful, particularly in how they re-imagine their discussion.

I found the discussion unhelpful. A more useful discussion would be a more in-depth interpretation of the included checklists/guidance. For example, do items overlap, are some checklists/guidance more helpful than others; should a new one be developed; is there merit in getting a the various groups involved and developed a more standardized checklist/guidance? The discussion is silent on whether the better guidance should be evidence based?

Was the search strategy peer reviewed (PRESS; McGowan J, Sampson M, Salzwedel DM, Cogo E, Foerster V, Lefebvre C. PRESS Peer Review of Electronic Search Strategies: 2015 Guideline Statement. J Clin Epidemiol. 2016 Jul;75:40-6). I also wonder whether the authors should have searched other sources, such as YouTube (we did this in the attached systematic review)? For example, is there merit on including Publons Academy? There may be merit (face validity) is examining potential resources that might exist by the International Committee of Medical Journal Editors; the World Association of Medical Editors; I’m not convinced excluding searching the EQUATOR library of reporting guidelines is reasonable. To be honest I do not know how many reporting guidelines make explicit mention of peer reviewers. I know that CONSORT explicitly mentions this “We developed CONSORT 2010 to assist authors in writing reports of randomized controlled trials, editors and peer reviewers in reviewing manuscripts for publication, and readers in critically appraising published articles.” as does the forthcoming PRISMA update “In order to achieve this, we encourage authors, editors and peer-reviewers to adopt the guideline”. It is possible to acknowledge this in the limitations of your scoping review.

6. PLOS authors have the option to publish the peer review history of their article (what does this mean?). If published, this will include your full peer review and any attached files.

Reviewer #1: **Yes: **Emily Ford

Reviewer #2: **Yes: **David Moher

---

## [Author Response · Author response to Decision Letter 0]

24 Mar 2021

We are grateful to the reviewers for their insightful comments and constructive suggestions. which greatly helped to improve our manuscript. Our point by point responses are as follows: 

Response to Reviewer #1:

1. There are several small changes you can make to the manuscript that will strengthen it. First, you make claims in your discussion section that I do not feel are warranted by the data you collected. It is not that these claims should not be considered, rather they simply need more research before we can make them. I suggest moving these to an expanded section discussing areas for further research.

Response) Thank you very much for your suggestion. We have revised our manuscript to discuss areas for further research. 

Abstract: “This calls for more research to determine the need to use uniform review standards for transparent and standardized peer review.”

Discussion: “Therefore, in order to improve current peer review conditions, we suggest that more research should be carried out and the various groups involved should come together to discuss about the need of a more standardized checklist to be developed in a systematic format and useful to many groups.”

2. It will also strengthen your article if you can provide more explicit scoping for the inclusion and exclusion criteria that you used in your study. For example, please clarify why you chose the websites that you listed, and why not other websites? (e.g. F1000Research, Peer J, eLife.) Additionally, if you can provide more detail about how those websites were searched.

Response) We have added more details on how the publishers were selected for screening. 

“For grey literature, websites of major biomedical and complementary medicine journal publishers including the following were searched. manually by browsing and clicking through the websites to look for guidelines for reviewers. The publishers were selected based on Simba Information (www.simbainformation.com)’s report, Global Medical Publishing 2019-2023, which provides a list of the leading medical publishers. More publishers were selected from those on Journal Citation Report (jcr.clarivate.com) in the Integrative and Complementary Medicine category. Subsidiary websites were not included in the search of major publishers.”

3. Your search strategies used are decent, though I do have questions about some of the searching decisions you made. Please make clear why you made certain decisions to search as you did and include that in your methods section.

Response) We have revised the search strategy part to make it clear of our search.

“In order to search more publications for comprehensive results, Text Word search was performed, and MeSH term for “peer review” was used for search in PubMed. The search strategy was reviewed by all authors and agreed upon.”

4. It would also strengthen your article if you expand upon the limitations of your study. I have outlined potential additions to the limitations on the uploaded pdf in the comments and changes. One example is that you might refer to the search strategy as a limitation, as well as the inclusion of articles only written in English.

Response) Thank you for your suggestion. We have expanded our study limitations relating to our search strategy and our inclusion of only English articles.

“While the manual search of websites for checklists added more to our data, this may have introduced human error. Other limitations were bias related to search strategies and inclusion criteria. Our preselected keyword search strategies may have introduced some bias or unintended exclusion of publications which should have been included. In addition, biomedical databases with access restrictions were also searched to provide more comprehensive data but this may cause barriers to the publications for researchers. The inclusion of only English language publications may have also excluded some content.”

5. Another way that you could strengthen this article would be to expand the conclusion and mention more specifically what other studies may come of your findings.

Response) We have revised our conclusion to mention more specifically about future studies.

“In order to produce objective criteria for reviewing manuscripts, further research is needed, and various groups involved in publishing academic manuscripts, such as publishers, editors, reviewers, and other experts, should come together to discuss on developing standard guidelines or recommendations for peer reviewers which may be used by itself or along with individual journal’s instructions for reviewers.”

6. There is also one more substantive change to consider—the nomenclature used to discuss articles. It may be much clearer to your readers if you refer to the data retrieved from databases as “publications” instead of “articles.” Given that “articles” are often used by databases as a publication type, using this word in the text of your manuscript to discuss the data can introduce confusion. If you were to use the word “publication” to mean those items found in databases, some of which *are* research articles, this would help eliminate some confusion.

Response) Thank you for bringing this to our attention. We have corrected the term “articles” to “publications” throughout the manuscript, to eliminate some confusion.

7. There are a number of minor changes that I think will greatly strengthen this article. As I have noted on the manuscript itself and in this document, please provide more details in your methods section regarding inclusion/exclusion, search strategies, etc. Please also expand the limitations section and areas for further study. Finally, please move conclusions unsubstantiated by the data into the areas for further study.

Response) Thank you very much for your comments. Your point by point comments on the PDF were very helpful in improving our paper. We have tried to revise all the points raised as we agree with you on all of them. Our methods section, limitations in our discussion section, and our conclusion have been revised as stated above.

Response to Reviewer #2:

1. While I think the scoping review provides some utility for readers, I think it needs more rigor and interpretation before it can be more meaningful to readers. 

Response) Thank you for your comments. We have revised our manuscript to add more interpretation and in-depth discussion to convey some meaningfulness.

2. A main concern of mine is with the output of the results. While the authors provide a long list of peer review guidance (Table 1), as a reader I’m lost. What usefulness to readers is an editorial with guidance criteria A,C,M,R,S (e.g., Alam, 2015)? If I go to Figure 2, I can try and unpack what this means. For example, I can see that A is ‘attitude’. But from a reader’s point of view what does this mean for peer reviewing? One way of solving this issue is to include a box with a clear glossary of explanation for each criterion in Figure 2 – this information should be included in the Methods section. The authors also need to describe how these criteria were selected and developed – please include this in the methods section as well. 

Response) Thank you for your suggestions. In Methods, we have included a glossary to make it clear about each peer review guidance criterion. We have also revised the manuscript to describe how the criteria were selected and developed: 

“The review guidance criteria were also developed together by the authors and agreed by all authors. The criteria were developed to group the contents of reviewer guidelines and to provide a clear overview of available guidelines for reviewers. Then for each included publication, the criteria were selected by two authors independently.”

3. The other aspect of Table 1 that concerns me is that it appears each row is equivalent to each other. There is no evidence-based hierarchy. For example, is an editorial with no checklist items as informative/evidence based as a guideline (e.g., Marusic, 2005). Should there no be a vetting and organization of all the guidance documents included? For this reason, I do wonder whether there is a need for the authors to complete some form of validity assessment of all of the included documents. You as authors, and readers more generally, need some way of separating the wheat from the chaff. 

Response) Thank you for bringing this to our attention and for the resource provided. In order to assess and organize the included publications in a better way, we have modified Table 1 and also modified a quality assessment tool from the study by Hawker et al. The assessment results and the tool used are provided as Supplement S4. Most of the included guidelines related publications were some form of editorial materials, and the included guidelines article was not the evidence-based guidelines developed by experts and consensus, although its article type was named ‘Guidelines.’

“In order to assess and organize the included publications, a modified quality assessment tool was modified from the tool suggested by Hawker et al.[11] and used in this study to assess the publications included.”

4. The results of the scoping review are also silent on whether there is overlap between the guidance documents. This point relates to the discussion. I have included a recent systematic review, also about checklists. The authors might find it helpful, particularly in how they re-imagine their discussion. I found the discussion unhelpful. A more useful discussion would be a more in-depth interpretation of the included checklists/guidance. For example, do items overlap, are some checklists/guidance more helpful than others; should a new one be developed; is there merit in getting the various groups involved and developed a more standardized checklist/guidance? The discussion is silent on whether the better guidance should be evidence based? 

Response) We agree that the better guidance should be evidence based. We have added our interpretation and elaborated further to discuss about the included checklist/guidance. 

“Primarily, checklists are meant to be mnemonic devices aimed at supporting authors and reviewers through a process with a general outline. The items relating to the evaluation of the manuscript structure or format had more overlaps than those relating to the evaluation of the manuscript content relating to methods and results, especially if the guidelines were intended for more specific groups or study designs. The general items of the review checklist in the biomedical related field are most likely to overlap with non-biomedical field as well. As the checklist can be designed based on the overall purpose, some checklist can be more useful than others. Therefore, in order to improve current peer review conditions, we suggest that the various groups involved to come together to discuss about the need of a more standardized checklist which should be developed in a systematic format.

Getting the various groups involved and developing a more standardized checklist would have a technical merit. Items employed for checklist should be from a broad range of academic groups and reliable sources. Various perspectives that could represent the reviewer committee in general should be considered and factored into the design of the checklist. This is particularly important if the checklist is designed to decrease errors of omission during peer review. Although reviewer checklists themselves are generally not included for publication even for open peer review, the development of reviewer guidelines should be evidence based and requires a systematic and comprehensive approach. The design of effective checklist should include utilization of published guidelines, formation of expert panels and extensive pilot-testing of preliminary checklists to ensure that recommendations are valid and reliable.”

5. Was the search strategy peer reviewed (PRESS; McGowan J, Sampson M, Salzwedel DM, Cogo E, Foerster V, Lefebvre C. PRESS Peer Review of Electronic Search Strategies: 2015 Guideline Statement. J Clin Epidemiol. 2016 Jul;75:40-6). I also wonder whether the authors should have searched other sources, such as YouTube (we did this in the attached systematic review)? For example, is there merit on including Publons Academy? There may be merit (face validity) is examining potential resources that might exist by the International Committee of Medical Journal Editors; the World Association of Medical Editors; I’m not convinced excluding searching the EQUATOR library of reporting guidelines is reasonable. To be honest I do not know how many reporting guidelines make explicit mention of peer reviewers. I know that CONSORT explicitly mentions this “We developed CONSORT 2010 to assist authors in writing reports of randomized controlled trials, editors and peer reviewers in reviewing manuscripts for publication, and readers in critically appraising published articles.” as does the forthcoming PRISMA update “In order to achieve this, we encourage authors, editors and peer-reviewers to adopt the guideline”. It is possible to acknowledge this in the limitations of your scoping review.

Response) Thank you for sharing the manuscript on Peer Review of Electronic Search Strategies (PRESS). In our future searches, we will refer to PRESS for peer reviewing our search strategy. For this scoping review, we did not have our search strategy reviewed externally but all authors reviewed the search strategy and agreed to it. We have stated this in our revised manuscript as the following: “The search strategy was reviewed by all authors and agreed upon.” 

Initially, the EQUATOR Network was excluded because they were intended primarily for authors. However, as you have pointed out, we agree that the purpose of the reporting guidelines is also to assist peer reviewers in reviewing manuscripts, so we have searched the EQUATOR Network and added the items from the reporting guidelines for main study types: CONSORT, STROBE, PRISMA, SPIRIT, STARD, CARE, SQUIRE, AGREE, ARRIVE, CHEERS, and SRQR. Thus 378 checklist items were added from searching the EQUATOR Network website. The methods section was revised as the following: “In addition, the websites of non-publishers were searched including Council of Science Editors (www.councilscienceeditors.org), the Committee on Publication Ethics (www.publicationethics.org), the EQUATOR Network (www.equator-network.org), and the ICMJE (www.icmje.org), the World Association of Medical Editors (www.wame.org), and Publons (www.publons.com).was also searched. For the EQUATOR reporting checklists published both on their website and some journals, they were considered as website items all under the EQUATOR Network as a non-publisher organization.”

We have also manually searched the ICMJE, the WAME, and the Publons websites and updated our paper to include their resources. However, we have not included YouTube in our hand search as we wanted to focus on printed or online published (PDF or html) form of checklist provided by reliable sources. Screening YouTube uploads could be challenging, but in the future studies, we will consider including YouTube.

---

## [Decision Letter · Decision Letter 1]

16 Apr 2021

PONE-D-20-36943R1

A scoping review on biomedical journal peer review guides for reviewers

PLOS ONE

Dear Dr. Ang,

Thank you for submitting your manuscript to PLOS ONE. After careful consideration, we feel that it has merit but does not fully meet PLOS ONE’s publication criteria as it currently stands. Therefore, we invite you to submit a revised version of the manuscript that addresses the points raised during the review process.

We look forward to receiving your revised manuscript.

Kind regards,

Tim Mathes

Academic Editor

PLOS ONE

Journal Requirements:

Reviewers' comments:

Reviewer's Responses to Questions

**Comments to the Author**

1. If the authors have adequately addressed your comments raised in a previous round of review and you feel that this manuscript is now acceptable for publication, you may indicate that here to bypass the “Comments to the Author” section, enter your conflict of interest statement in the “Confidential to Editor” section, and submit your "Accept" recommendation.

Reviewer #1: All comments have been addressed

Reviewer #2: (No Response)

2. Is the manuscript technically sound, and do the data support the conclusions?

Reviewer #1: Yes

Reviewer #2: Yes

3. Has the statistical analysis been performed appropriately and rigorously? 

Reviewer #1: Yes

Reviewer #2: N/A

4. Have the authors made all data underlying the findings in their manuscript fully available?

Reviewer #1: Yes

Reviewer #2: No

5. Is the manuscript presented in an intelligible fashion and written in standard English?

Reviewer #1: Yes

Reviewer #2: Yes

6. Review Comments to the Author

Reviewer #1: I am so pleased to see that the authors have made marked improvements to this manuscript based on both referees' suggestions. The additional language about method and search strategy provides more transparency to potential limitations of the study as well as reproducibility. Moreover, the expanded discussion section and areas for further research clearly outline what more can be done to move in the direction for providing better guidance for referees.

One thing I noticed that should be corrected prior to publication, is the renaming of the "article type" column in Table 1. I think it should be relabeled to be "publication type."

Reviewer #2: Thank you to the authors for reading my peer review comments. I appreciated the responses and subsequent revision of the manuscript.

I think the paper is improved; well done. I have a few additional comments.

1. I did forget to ask the authors as to whether they searched the website of the International Congress on Peer Review and Scientific Publication (https://peerreviewcongress.org/ ). Even as a measure of face validity, it would be reasonable to search. I assume the search strategy used by the authors picked up the journal Research Integrity and Peer review (https://researchintegrityjournal.biomedcentral.com/ )?

2. In the Methods section – 2.4 Outcomes, the authors state “The criteria were developed together by the authors and agreed by all authors”. I appreciate the criteria were developed together by the authors. I think this requires some unpacking. Was this based on anecdotal experiences of the authors as peer reviewers; evidence-based publications to inform the criteria or some other way? My point here is that a reader interested in replicating your methods needs details.

3. Please consider adding this randomized trial (protocol) in the discussion (Speich B, Schroter S, Briel M, Moher D, Puebla I, Clark A, Maia Schlüssel M, Ravaud P, Boutron I, Hopewell S. Impact of a short version of the CONSORT checklist for peer reviewers to improve the reporting of randomised controlled trials published in biomedical journals: study protocol for a randomised controlled trial. BMJ Open. 2020 Mar 19;10(3):e035114. doi: 10.1136/bmjopen-2019-035114. PMID: 32198306; PMCID: PMC7103787). It is an example of a standardized checklist for peer reviewing randomized trials.

7. PLOS authors have the option to publish the peer review history of their article (what does this mean?). If published, this will include your full peer review and any attached files.

Reviewer #1: **Yes: **Emily Ford, Associate Professor, Urban & Public Affairs Librarian, Portland State University

Reviewer #2: **Yes: **David Moher

---

## [Author Response · Author response to Decision Letter 1]

26 Apr 2021

Once again, we are grateful to the reviewers for their additional comments and suggestions to better improve our manuscript. Our point by point responses are as follows: 

Reviewer #1: 

1. One thing I noticed that should be corrected prior to publication, is the renaming of the "article type" column in Table 1. I think it should be relabeled to be "publication type."

Thank you for pointing the missed-out correction. We have now corrected it to “publication type.”

Reviewer #2: 

1. I did forget to ask the authors as to whether they searched the website of the International Congress on Peer Review and Scientific Publication (https://peerreviewcongress.org/ ). Even as a measure of face validity, it would be reasonable to search. I assume the search strategy used by the authors picked up the journal Research Integrity and Peer review (https://researchintegrityjournal.biomedcentral.com/ )? 

Thank you for your comments. We have searched peerreviewcongress.org. 

For Research Integrity and Peer Review (journal), the journal is on PubMed so they were searched together with others when we did the search.

2. In the Methods section – 2.4 Outcomes, the authors state “The criteria were developed together by the authors and agreed by all authors”. I appreciate the criteria were developed together by the authors. I think this requires some unpacking. Was this based on anecdotal experiences of the authors as peer reviewers; evidence-based publications to inform the criteria or some other way? My point here is that a reader interested in replicating your methods needs details. 

Thank you for bringing this to our attention. We have revised the sentence to provide more details for readers: 

“Before going through the screening, after the search of literature, the authors randomly pre-screened 10 publications to look for possible review guidance criteria, and based on the discussion by all authors with experience as peer reviewers, the criteria were developed.” 

3. Please consider adding this randomized trial (protocol) in the discussion (Speich B, Schroter S, Briel M, Moher D, Puebla I, Clark A, Maia Schlüssel M, Ravaud P, Boutron I, Hopewell S. Impact of a short version of the CONSORT checklist for peer reviewers to improve the reporting of randomised controlled trials published in biomedical journals: study protocol for a randomised controlled trial. BMJ Open. 2020 Mar 19;10(3):e035114. doi: 10.1136/bmjopen-2019-035114. PMID: 32198306; PMCID: PMC7103787). It is an example of a standardized checklist for peer reviewing randomized trials.

Thank you for suggesting an additional relevant paper which would enrich our discussion further. It is an important study and add to the evidence. We have added the protocol of the randomized controlled trial. We have added the below text in our discussion:

“This kind of attempt is still in progress as seen in a study protocol of randomized controlled trial by Speich et al.[87]. This trial aims to evaluate the effectiveness or the need of providing a short version of the CONSORT checklist and its explanation to peer reviewers for them to check during peer review.”

---

## [Editor Report · Decision Letter 2]

27 Apr 2021

A scoping review on biomedical journal peer review guides for reviewers

PONE-D-20-36943R2

Dear Dr. Ang,

We’re pleased to inform you that your manuscript has been judged scientifically suitable for publication and will be formally accepted for publication once it meets all outstanding technical requirements.

Kind regards,

Tim Mathes

Academic Editor

PLOS ONE
---

## [Editor Report · Acceptance letter]

11 May 2021

PONE-D-20-36943R2 

A scoping review on biomedical journal peer review guides for reviewers 

Dear Dr. Ang:

I'm pleased to inform you that your manuscript has been deemed suitable for publication in PLOS ONE. Congratulations! Your manuscript is now with our production department. 

Kind regards, 

on behalf of

Dr. Tim Mathes 

Academic Editor

PLOS ONE